# Diversification is All You Need: Towards Data Efficient Image Understanding

**Abstract.** One of the issues one comes across while dealing with image understanding problems, such as image classification and semantic segmentation, is the lack of enough number of labeled images in the training set which often results in overfitting. To deal with this issue, we proposed to diversify the models, the data and the test samples to gain the competitive performance.

Specifically, for image classification, we adopt a two-stage framework to train the network. At the first stage, we train several different models independently from each other while varying the backbone architecture and input modality by combining two SOTA data augmentation techniques Augmix [3] and Mixup [12]. At the second stage, we do ensemble classification, in which we combine the set of trained models to classify unseen image rather than just a single one. In the experiments on the subset of Imagenet dataset, our method consistently improves accuracy from the baseline for the test samples in the dataset. For semantic segmentation, we proposed the seg-Augmix that extended the Augmix algorithm to the semantic segmentation task. In addition, the Frequency Weighted model ensemble method is also proposed to improve the performance when combining different models. Using the proposed method we are able to achieve competitive performance on the Semantic Segmentation track and Image classification track of the VIPriors 2020 challenge[1] respectively.

**Keywords:** Image Classification, Semantic Segmentation, Data Augmentation, Augmix, Mixup, seg-Augmix, Frequency weighted model ensemble, Test Time Augmentation

## 1 Introduction

Deep neural networks (**DNNs**) excel at image understanding tasks, such as image classification and semantic segmentation, when labeled data are abundant, yet their performance degrades substantially when provided with limited supervision. Improving the generalization ability of these models in low data regimes is one of the most difficult challenges. Data efficient image understanding is an important topic but often has been ignored in the computer vision community

---

[1] https://vipriors.github.io/challenges/

until recently. Methods like [5] are proposed to utilize prior translation invariance knowledge to address the small dataset challenge.

Data Augmentation (**DA**) [6] is another very powerful method to address this challenge. The augmented data will represent a more comprehensive set of possible data points, thus minimizing the distance between the training and any future testing sets. **DA** approaches overfitting from the root of the problem, the training dataset. This is done under the assumption that more information can be extracted from the original dataset through augmentations. These augmentations artificially inflate the training dataset size by transforming existing images such that their label is preserved. This encompasses augmentations such as geometric and color transformations, random erasing, adversarial training, neural style transfer, etc. Another type of augmentation methods called oversampling augmentations create synthetic instances and add them to the training set. This includes mixing images, feature space augmentations, and generative adversarial networks (**GANs**), etc. In this paper, we proposed to diversify the models, the data and the test samples for data efficient image understanding problem for image classification and semantic segmentation tasks.

Specifically, the image classification system must overcome issues of viewpoint, lighting, occlusion, background, scale, etc. The task of **DA** is to incorporate such invariances into the dataset such that the resulting models will perform well despite these challenges. In this paper, our proposed method can be characterized as Augmix [3] combined with Mixup [12] with EfficientNet backbone. In Section 2 we present a brief description of these **DA** methods, and EfficientNet architecture. Next, we will discuss our contribution for the image classification task. General scheme of our approach for image classification is shown in Figure 3.1.

Semantic segmentation is the task of predicting the class of each pixel in the image. This paper focuses on the scenarios where the labeled semantic segmentation training dataset is very small. Recently several works have been done to address the issue of lack of training data by data augmentation for image classification, such as Augmix, Mixup etc. However, it is not trivial to extend those works to the area of semantic segmentation since it is a position sensitive task that needs careful consideration of changing the labels when image is augmented. In this paper, we proposed the seg-Augmix that extended the Augmix algorithm to the semantic segmentation task. Moreover, the Frequency Weighted (FW) model ensemble method is also proposed to improve the performance when combining different models.

Our major contributions are summarized as follows.

1. We propose to combine Augmix and mixup to employ both the within-class diversification and between-class diversification for image classification task.

2. We propose seg-Augmix that extends Augmix method to semantic segmentation.

3. we propose Frequency weighted model ensemble method to utilize the properties of different models for semantic segmentation task.

## 2    Preliminaries

### 2.1    EfficientNet

Many methods for increasing deep learning generalization performance focus on the model's architecture itself. This has led to a sequence of progressively more complex architectures from AlexNet [10] to VGG-16 [8], ResNet [2], DenseNet [4], etc.

For the image classification task, we decided to use state-of-the-art ImageNet classification architecture EfficientNet [4]. This model is a result of neural architecture search with carefully balancing of network depth, width and resolution. It is also shown that better resulting models from image classification and transfer learning have even less number of parameters.

### 2.2    Augmix

Augmix is a novel **DA** method which improves the accuracy of the network for several specific shifted domain scenarios. The main goal of Augmix is to increase the robustness of the deep model trained on the augmented data to generalize well beyond the data corruption like the rotation, translation, noise, etc. For each input, they apply different operation of image shift and make the weighted combination of them. The weight vector is generated randomly from Dirichlet distribution [3]. The weighted combined images would be added to the original image in convex combination. The convex weights are generated from Beta distribution.

**Augmentations**: Augmix consists of mixing the results from compositions of augmentation operations (it uses the operations from AutoAugment [3]). Next, it randomly sample k augmentation chains, where k = 3 by default. Each augmentation chain is constructed by composing from one to three randomly selected augmentation operations.

**Mixing**: The resulting images from these augmentation chains are combined by mixing. The k-dimensional vector of convex coefficients is randomly sampled from a Dirichlet distribution with parameter $(\alpha, ..., \alpha)$. Once these images are mixed, the result of the augmentation chain and the original image are combined through another random convex combination sampled from a Beta distribution with parameter $(\alpha, \alpha)$.

Later they train the network with adding the Jensen-Shannon divergence for the posterior distributions of augmented images as the consistency regularizer [3]. They show this data augmentation will increase the accuracy of the model for shifted and non-shifted domains and also it leads to more calibrated model for domain shift problem.

### 2.3    Mixup

Mixup [12] is a recently proposed technique for training **DNNs** where additional data points are generated during training using convex combination of random

pairs of images and their associated labels. While simple to implement, mixup is a surprisingly effective method of data augmentation for image classification: **DNNs** trained with mixup show noticeable improvements in classification performance on a number of image classification benchmarks.

Mixup training is based on the principle of Vicinal Risk Minimization (**VRM**) [1]: the classifier is trained not only on the training samples, but also in the vicinity of each training data point. In mixup, the vicinal points are generated according to the following simple rule:

$$\tilde{x} = \lambda x_i + (1 - \lambda)x_j, \quad \tilde{y} = \lambda y_i + (1 - \lambda)y_j$$

where $x_i$ and $x_j$ are two data points randomly sampled from the training set, $y_i$ and $y_j$ are their associated one-hot encoded labels, and $\lambda \in [0, 1]$ is the mixing coefficient. In other words, mixup extends the training distribution by incorporating the prior knowledge that linear interpolations of feature vectors should lead to linear interpolations of the associated targets.

One intuition behind this is that by linearly interpolating between samples, we encourage the classifier to act smoothly and kind of interpolate nicely between samples without sharp transitions.

### 2.4   HRNet and OCR

For the semantic segmentation task, the key idea of HRNetV2 [9] is to maintain the high resolution of the feature map since dense pixel prediction tasks such as semantic segmentation, and depth estimation, will benefit the higher resolution of the feature map. Meanwhile, the multiple scale feature fusion is another important aspect of HRNetV2 which improves the performance of semantic segmentation tasks. In addition to fusing the multiple scale features at the end of the backbone, HRNetV2 also fuse multiple scales in the middle of the backbone whenever down-sampling happens [9].

Object Contextual Representation (OCR) [11] is applied after HRNetV2 to model the context information for per pixel classification. The key idea of OCR is to model the object level context information for each pixel. Specifically the object level context of each pixel is defined as a weighted combination of object region feature, whose weights are determined by the similarities between the feature at the pixel and the feature of object region.

## 3    Proposed Method

### 3.1   Data Diversification

Deep Convolutional Neural Networks benefits a lot when labeled training data are abundant, yet their performance degrades substantially when provided with limited supervision. Improving the generalization ability of these models in small size training data regimes is one of the most difficult challenge. Models with poor generalizability will overfit the training data. Data Augmentation is a very

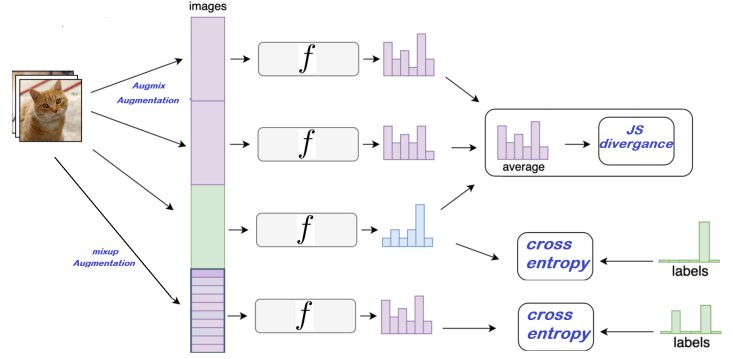

**Fig. 1.** The proposed data diversification scheme for Image classification.

powerful method to address this challenge. The augmented data will represent a more comprehensive set of possible data points, thus minimizing the distance between the training and any future testing sets.

To achieve this goal, we propose a combination of Augmix and Mixup for image classification, and seg-Augmix, an extension of Augmix, for semantic segmentation. The goal of combining Augmix and Mixup for image classification is to combine the benefits of both approaches as Augmix focuses on within-class data diversification, while Mixup benefits from the between-class data diversification. We also propose the seg-augmix and its goal is to generate the augmented data for semantic segmentation without changing the positions of objects in the image.

**Proposed Data Augmentation for Image Classification** Without loss of generality, we consider a multi-class ($K$ class) classification problem as the running task example.

Consider the joint space of inputs and class labels, $\mathcal{X} \times \mathcal{Y}$ where $\mathcal{X} = \mathbb{R}^d$ and $\mathcal{Y} = \{1, ..., K\}$ for ($K$way) classification, Let $\mathcal{P}_{x \times y}$ be the probability distribution of the data points on these joint space. Inspired from Augmix [3], and mixup [12] methods, our goal is to learn a classifier $f_\theta : \mathcal{X} \to \mathcal{Y}$ with parameter $\theta$, using the following proposed optimization problem:

$$\theta^* = \arg \min_{\theta} \mathbb{E}_{(x,y) \sim \mathcal{P}_{x \times y}} \big[ \mathcal{L}_{cls}(f_\theta(x), y) + \gamma \, \mathcal{L}_{js}(f_\theta(x), f_\theta(x'), f_\theta(x'')) \big] +$$

$$\beta \, \mathbb{E}_{(x_1,y_1) \sim \mathcal{P}_{x \times y}} \mathbb{E}_{(x_2,y_2) \sim \mathcal{P}_{x \times y}} \mathbb{E}_{\lambda \sim Beta(\alpha)} \big[ \mathcal{L}_{cls}(f_\theta(\lambda x_1 + (1-\lambda)x_2), \lambda y_1 + (1-\lambda)y_2) \big]$$

$$(1)$$

where $\mathbb{E}$ denote the expectation operator, $\mathcal{L}_{cls}$ denotes the standard cross entropy loss, $x'$ and $x''$ are two augmentations of $x$ using the Augmix method [3], $\alpha$, $\beta$, and $\gamma$ are the method's hyperparameters, and $\mathcal{L}_{js}$ is the Jensen-Shannon divergence between the classifier output of the original sample $x$ and its augmentations $x'$ and $x''$.

Since the semantic content of an image is approximately preserved with Aug-mix augmentation, with $\mathcal{L}_{js}$, we encourage the classifier $f$ to map $x$, $x'$, and $x''$ close to each other in the output space. This is done by first obtaining $\mathcal{M} = \big(f_\theta(x) + f_\theta(x') + f_\theta(x'')\big)/3$, and then computing

$$\mathcal{L}_{js}(f_\theta(x), f_\theta(x'), f_\theta(x'')) = \frac{1}{3}\bigg( KL[f_\theta(x); \mathcal{M}] + KL[f_\theta(x'); \mathcal{M}] + KL[f_\theta(x''); \mathcal{M}]\bigg) \tag{2}$$

where $KL[p; q]$ denotes the KL divergence between two probability vector $p$ and $q$ (it should be noted that the output of the classifier $f$ is a $C$ dimensional prob-ability vector).

We solve the above optimization problem with Stochastic Gradient Descent (SGD) by approximating the expectation with the sample averages.

**Proposed Data Augmentation for Semantic Segmentation** We proposed a novel seg-Augmix data augmentation method that extended the convention Augmix method [3] to the semantic segmentation task, in order to deal with small number of labeled training examples. The following steps are conducted when applying the seg-Augmix method.

1. The proposed seg-Augmix will first maintain a pool of data augmentation techniques that will not change the position of each region so that the seg-mentation map will not be changed. It will ease the training and avoid po-tential error of changing the segmentation map to restrict the data augmen-tation pool.
2. Then, several randomly chosen data augmentation methods from the pool will be applied to each batch during the training to generate the augmented images without changing the labels. Those augmented images will be mixed with the original image.
3. Finally, the Jensen-Shannon Divergence Consistency is applied on the origi-nal images and the augmented images to prevent the instability of the train-ing.

Specifically, in step (1), the data augmentation pool consists of autocontrast, equalize, posterize, solarize, color, contrast, brightness, and sharpness data aug-mentation techniques, and None operation (Figure 2) that means no data aug-mentation is applied. Those methods only manipulate the pixel values rather than changing the position of the regions so that the ground truth segmenta-tion map will be intact. In step (2), two hyper-parameters, namely the mixture width $M_w$, and mixture depth $M_d$, will be defined first. Mixture width defines the number of branches for generating the augmented images, and mixture depth defines the number of consecutive data augmentations. As an example illustrated in Figure 2, mixture width $M_w=3$ and mixture depth $M_d=3$. We denote $f_i^{M_d}(x)$ as applying the randomly picked data augmentation function $f$ from data aug-mentation pool for $M_d$ times sequentially for branch $i$. Please also note that the "None" in the data augmentation pool might also be chosen inside each branch, shown as the dashed orange box in Figure 2.

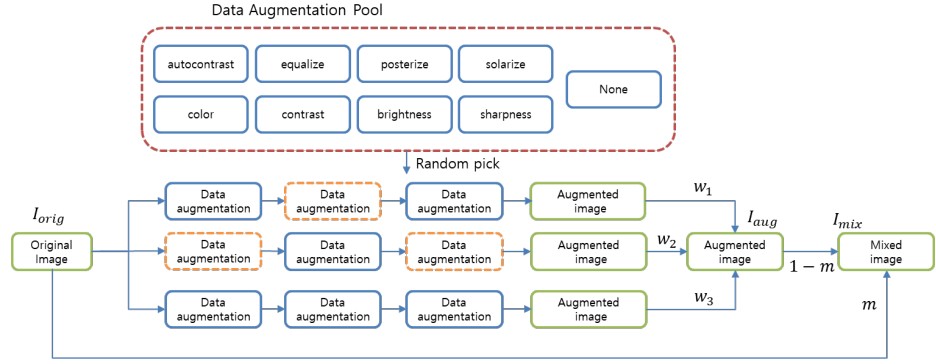

**Fig. 2.** The detailed steps of seg-Augmix. Each "data augmentation" is randomly picked from the data augmentation pool. mixture width $M_w = 3$ and mixture depth $M_d = 3$ is used in this example, which defined the number of branches and the number of consecutive data augmentations, respectively. For each branch, the mixture weights are randomly generated from Dirichlet distribution. The final combination with the original image used the weights generated from Beta distribution. Please also note that the "None" in the data augmentation pool means no data augmentation will be applied, and the dashed orange box means the "None" is picked.

For each branch $i$, the mixture weights $w_i$ are randomly generated from Dirichlet distribution. The final combination with the original image used the weights m generated from Beta distribution. Mathematically, the generated augmented image $I_{aug}$, and the generated mixed image $I_{mix}$ will be defined as follows.

$$I_{aug} = \sum_{i=1}^{M_w} w_i f_i^{M_d}(I_{orig}), j \in 1, 2, \ldots, M_d \tag{3}$$

$$I_{mix} = (1 - m) * I_{aug} + m * I_{orig} \tag{4}$$

In step (3), we will generate two mixed images, and feed the original image, and two mixed images into the network to generate three softmax logits, $p_o$, $p_{m1}$, and $p_{m2}$, respectively. Then the additional Jensen-Shannon Divergence Consistency is defined as follows. $KL[x||y]$ defines the KL divergence between $x$ and $y$.

$$JS(p_o, p_{m1}, p_{m2}) = \frac{1}{3}(KL[p_o||M] + KL[p_{m1}||M] + KL[p_{m2}||M]) \tag{5}$$

$$M = \frac{1}{3}(p_o + p_{m1} + p_{m2}) \tag{6}$$

Figure 3 also shows some examples to show how the seg-Augmix works.

### 3.2 Test Diversification

**Image Classification** Test-time augmentation, or **TTA** for short, is an application of data augmentation to the test dataset. Specifically, it involves creating

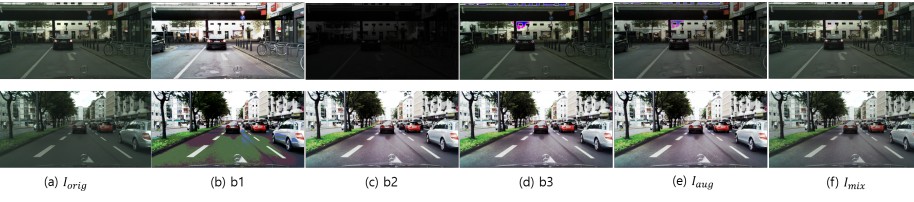

(a) $I_{orig}$        (b) b1        (c) b2        (d) b3        (e) $I_{aug}$        (f) $I_{mix}$

**Fig. 3.** Examples of images generated after applying seg-Augmix. (a) Original image, (b)-(d) Generatead images for each branch, (e) Augmented image after combining images from (b)-(d), and (f) Final mixed image. Better viewed in color and enlarged.

multiple augmented copies of each image in the test set, having the model make a prediction for each, then returning an ensemble of those predictions.

In this work, a single simple test-time augmentation is performed, by randomly croping the test image 10 times, make a prediction for each, then returning an ensemble average of those predictions.

**Semantic Segmentation** we first applied the widely used test time augmentation techniques. Specifically, it includes the multiple scale, and horizontal flip augmentation.

We applied Frequency Weighted (FW) model ensemble method instead of average model ensemble method to improve the performance.

1. Based on the observation that the training dataset is imbalanced, we train an Online Hard Example Mining (OHEM) [7] based version for each model.
2. The per class performance indicates that OHEM model performs better at low frequency classes, while the original model performs well at higher frequency models. Besides, the seg-Augmix based model can also improve the performance of low frequent classes because of applying data augmentation. We also applied higher weights to those classes.
3. As a result, we will assign more weights for low frequency classes for output logits for OHEM model and seg-Augmix model, when ensemble them with the output logit of original model.

Specifically, for the OHEM and seg-Augmix models, the logits are combined such that the low frequency classes will have higher weights. Assume the output softmax logit of the $i$-th model is $\mathbf{L_i} \in R^{H \times W \times C}, i = 1, 2, ..K$, $\mathbf{w_i} \in R^{H \times W \times C}$ is the per class weights for all the C classes, where the values in each channel of $w_i$ are the same, then the predicted segmentation map of model ensemble will be as follows given than $argmax(x, axis)$ will take the argmax of $x$ along the axis.

$$seg = argmax(\sum_i \mathbf{L_i} \odot \mathbf{w_i}, axis = -1) \tag{7}$$

### 3.3   Model Diversification

**Image Classification** It is widely known that the ensemble averaging of neural networks trained independently leads to the improvement of test accuracy. In this work we trained models with different EfficientNet backbones and used equal average of predictions from these models to make final prediction.

**Semantic Segmentation** Conventional semantic segmentation models applied the encoder and decoder architecture, and the input feature size will be reduced 1/16 after the encoder. The loss of information caused by the 1/16 downsampling can be compensated if there is enough labeled training data.

However, in the data efficient semantic segmentation regime, where very limited labeled training dataset is available, we often prefer to keep as much information as possible after the encoder rather than the aggressive 1/16 downsampling. As a result, we propose to use the HRNetV2 [9] and Object Contextual Representation [11] as our base model for semantic segmentation task.

Several HRNetV2 variants have been developed for scenarios with different computational cost. For the experiments, we adopted two models, namely the HRNetV2-W48, and HRNetV2-W44, where 48 and 44 are the channel sizes of the highest resolution feature map after the HRNetV2 encoder.

## 4   Experimental Results

In this section, we explain the details of the experimental results for the image classification and semantic segmentation tasks.

### 4.1   Image Classification

We compare our approach to several baseline models on the 1000-class ImageNet 2012 dataset. More specifically, we pick 100 random samples per class from the Imagenet training set as the training set (100,000 samples in total). For the sake of ablation study, in this paper (and not for challenge submission), we constructed a test dataset by randomly picking 50 samples per class from the Imagenet validation set.

**Implementation Details** We implemented our method and baseline models in Pytorch. We used open source EfficientNet models (B0-B7)provided at https://github.com/rwightman/pytorch-image-models/. We trained all the networks with RMSProp optimizer. At first, all the model parameters are initialized randomly with normal distribution. We set the regularization parameters $\alpha = 0.2$, $\beta = 1$, and $\gamma = 12$ for all experiments. We trained all the models for 450 epochs. The details of the hyper-parameter settings for the EfficientNet models is available in Table 1.

**Table 1.** Hyper-parameter settings for different models.

| model | EfficientNet-B0 | EfficientNet-B1 | EfficientNet-B2 | EfficientNet-B3 | EfficientNet-B4 | EfficientNet-B5 | EfficientNet-B6 | EfficientNet-B7 |
|---|---|---|---|---|---|---|---|---|
| Batch size | 32 | 32 | 32 | 32 | 32 | 32 | 16 | 16 |
| Crop size | 224 | 224 | 260 | 260 | 260 | 260 | 260 | 224 |
| # epochs | 450 | 450 | 450 | 450 | 450 | 450 | 450 | 450 |
| Dropout rate | 0.2 | 0.3 | 0.3 | 0.3 | 0.3 | 0.3 | 0.3 | 0.3 |
| Drop connect rate | 0.2 | 0.2 | 0.2 | 0.2 | 0.2 | 0.2 | 0.2 | 0.2 |
| Weight Decay | 0.00001 | 0.00001 | 0.00001 | 0.00001 | 0.00001 | 0.00001 | 0.00001 | 0.00001 |
| Warm up Learning rate | 0.000001 | 0.000001 | 0.000001 | 0.000001 | 0.000001 | 0.000001 | 0.000001 | 0.000001 |
| Learning rate | 0.048 | 0.016 | 0.016 | 0.016 | 0.016 | 0.016 | 0.016 | 0.016 |

**Table 2.** Classification accuracy results (%) for different models (PM stands for Proposed Method).

| model | EfficientNet-B0 | EfficientNet-B1 | EfficientNet-B2 | EfficientNet-B3 | EfficientNet-B4 | EfficientNet-B5 | EfficientNet-B6 | EfficientNet-B7 |
|---|---|---|---|---|---|---|---|---|
| Baseline | 49.23 | 49.66 | 49.73 | 50.55 | 51.06 | 50.90 | 51.00 | 51.10 |
| Mixup-augment | 51.41 | 52.36 | 52.18 | 53.56 | 53.01 | 53.86 | 53.91 | 53.80 |
| Augmix-augment | 56.29 | 56.87 | 56.08 | 56.95 | 56.77 | 56.80 | 56.44 | 56.70 |
| **PM** | **59.60** | **61.80** | **60.60** | **62.96** | **62.04** | **63.07** | **63.03** | **63.06** |

**VIPrior Challenge Results** For this challenge, we use 50,000 labeled training samples and 50,000 labeled validation set for training the networks provided by the challenge organizers, and tested our model on the test set of the challenge (50,000 random samples from the validation set of the Imagenet2012 dataset). The submitted results for the challenge is the ensemble average of the models **EfficientNet-B1, EfficientNet-B3, EfficientNet-B5, EfficientNet-B6, EfficientNet-B7** and the obtained accuracy based on the colab evaluation was **66.39%**.

**Emperical Results** We compared our method with three baselines: (i) No-Augment: using the cross entropy loss only on the original data without any data augmentation. (ii)Augmix-augment: augmenting data using augmix method and applying the augmix loss on the augmented samples. (iii) mixup-augment: augmenting data using mixup method and applying the cross entropy loss on the augmented samples. The classification accuracy results for different methods is shown in Tables 2, and 3. As can be seen, from table 2, just training the plain EfficientNet models wont produce good results (first row) due to the small number of training samples. On the other hand, the Augmix and mixup methods which uses data augmentation, can improve the performance by 5% and 2.5% in average respectively. Moreover, our proposed method has better performance than Augmix and mixup due to the fact that it could produce more diverse augmented samples by jointly training the network with Augmix and Mixup.

In table 3, we showed the results after model ensemble. In the first row, we sorted the results of different EfficientNet models. In the second row, we provided the results of our ensemble-combining method by adding one model at a time to the ensemble. As can be seen, when we combine EfficientNet-B5 and EfficientNet-B6, we get 65.45% accuracy which is 2.4% better than EfficientNet-B5.

**Table 3.** Classification accuracy results (%) with the model ensemble. Each column in the second row shows the results for the ensemble of the models in the previous columns.

| model | EfficientNet-B5 | EfficientNet-B6 | EfficientNet-B7 | EfficientNet-B3 | EfficientNet-B1 | EfficientNet-B4 | EfficientNet-B2 | EfficientNet-B0 |
|---|---|---|---|---|---|---|---|---|
| PM | 63.07 | 63.06 | 63.03 | 62.96 | 62.04 | 61.80 | 60.60 | 59.60 |
| PM with Ensembeling | 63.07 | 65.05 | 65.64 | 66.10 | 66.02 | 66.01 | 65.89 | 65.55 |

**Table 4.** Model sensitivity to $\alpha$.

| model | $\alpha = 0$ | $\alpha = 0.2$ | $\alpha = 0.4$ | $\alpha = 0.6$ | $\alpha = 0.8$ | $\alpha = 1.0$ |
|---|---|---|---|---|---|---|
| **EfficientNet-B0** | 56.01 | **59.60** | 58.20 | 57.10 | 55.40 | 55.02 |
| **EfficientNet-B1** | 56.80 | 61.80 | 60.14 | 60.01 | **65.09** | 64.14 |

**Ablation Study** In the experiments above, we keep $\alpha = 0.2, \beta = 1, \gamma = 12$. To analyze the sensitivity of our method to changes in the aforementioned parameters, we conducted additional experiments on architectures **EfficientNet-B0** and **EfficientNet-B1** to analyze the parameter sensitivity of our method w.r.t. the various values of $\alpha, \beta$ and $\gamma$.

Tables 4, 5, and 6 shows the sensitivity analysis for the parameters of our model on the classification performance. Sensitivity analysis is performed by varying one parameter at the time over a given range, while for the other parameters we set them to their final values $\alpha = 0.2, \beta = 1, \gamma = 12$. From Tab. 4, we see that when $\alpha = 0$ (no mixup augmentation is considered), our method is equivalent to **Augmix-augment** method in which only the augmix augmentation is applied to the data points that leads to non-optimal solution. For high values of $\alpha$ (near 1), $\lambda$ tend to be close to 0.5 that means the samples are mix with equal weights. This could be problematic as the mixed samples could be very confusing (over-smoothing) for the classifier that leads to lower classification performance.

From Tab. 5, we see that when $\beta = 0$, we simply remove the mixup augmentation form the model that leads to inferiror performance. For other values of $\beta$, the performance is superior which demonstrates the effectiveness of the mixup in combination with augmix method.

Similarly, from Tab. 6, when $\gamma = 0$, we ignore the JS divergence loss term that leads to drop in the model's performance. This demonstrates that the smoothing out the classifier to be insensitive to the small perturbation of the data points plays a critical role in the classification task. For other values of $\gamma$, the performance is superior and there is little variation in the model performance, evidencing the robustness of the model w.r.t. $\gamma$.

### 4.2  Semantic Segmentation

**Dataset** The VIPriors Challenge Semantic Segmentation track provided a subset of the Cityscapes dataset, namely the MiniCity dataset. More specifically, it included training, validation and testing sets of 200, 100 and 200 images, respectively. In comparison with the original Cityscapes dataset, the size of the

**Table 5.** Model sensitivity to $\beta$.

| model | $\beta = 0$ | $\beta = 0.01$ | $\beta = 0.1$ | $\beta = 1$ | $\beta = 10$ | $\beta = 100$ |
|---|---|---|---|---|---|---|
| **EfficientNet-B0** | 56.29 | 58.10 | 59.30 | **59.60** | 59.00 | 58.60 |
| **EfficientNet-B1** | 56.87 | 60.00 | 60.24 | **60.60** | 60.10 | 59.80 |

**Table 6.** Model sensitivity to $\gamma$.

| model | $\gamma = 0$ | $\gamma = 1$ | $\gamma = 5$ | $\gamma = 10$ | $\gamma = 12$ | $\gamma = 20$ |
|---|---|---|---|---|---|---|
| **EfficientNet-B0** | 55.20 | 59.00 | 59.20 | 59.40 | **59.60** | 59.51 |
| **EfficientNet-B1** | 55.32 | 61.00 | 61.20 | 61.41 | **61.80** | 61.54 |

training dataset of MiniCity is significantly smaller, making the training the semantic segmentation networks very challenging.

**Implementation Details** We implemented our methods based on the open source code of HRNetV2, and Augmix Github repository. All the models are obtained by three stages of training. At first, all the model parameters are initialized randomly with normal distribution. The initial learning rate is 0.01, then the learning rate is decayed according to the **poly** leaning rate policy, where the learning rate is multiplied by $1 - (\frac{\text{iter}}{\text{max\_iter}})^{\text{power}}$ with power $= 0.9$. The weight decay is 5e-4, the batch size is 24, and the number of epochs is 4840. For the loss function, both softmax cross entropy loss function and the Jensen-Shannon Divergence Consistency loss used the same weights 1.0. We also applied random cropping with crop size $512 \times 1024$, random scaling in the range of [0.5, 2.0] and step size 0.25, random horizontal flip, as well as random color jittering including brightness, contrast, saturation and hue jittering as additional data augmentation methods.

The second stage is refinement stage, where several iterations of refinement is applied until the performance on validation dataset is saturated. For each iteration of refinement, a lower learning rate 0.001 is applied and the model is initialized with the best checkpoint at the first stage. The motivation of refinement stage is that the base training might not be stable since the training size is

| Method | mIoU (%) validation dataset |
|---|---|
| HRNetV2-W48 + OCR | 59.21 |
| HRNetV2-W48 + OCR + OHEM | 59.25 |
| HRNetV2-W48 + OCR + seg-Augmix | 59.84 |
| Average ensemble of above | 59.85 |
| FW model ensemble | 60.60 |

**Table 7.** The performance of HRNetV2-W48 + OCR, HRNetV2-W48 + OCR + OHEM, HRNetV2-W48 + OCR + seg-Augmix models, average model ensemble of them, and Frequency Weighted model ensemble of them on validation dataset of MiniCity.

| Method | mIoU (%) MiniCity Test dataset |
|---|---|
| **Our Entry** | 65.6 |

**Table 8.** The performance of our entry in the Semantic Segmentation track of the challenge.

very small and several iterations of refinement can obtain better performance. In our experiments, the HRNetV2-W48 + OCR + seg-Augmix model can achieve 57.71% mIoU at the base learning stage, and achieve better performance 59.84% after the refinement stage. Please note that the OHEM model is obtained from the refinement stage.

The third stage is finetuning stage that is applied for submission to the challenge only, where the validation dataset is also included in the training with a lower learning rate 0.001 for finetuning for 484 epochs. At this stage, the model is initialized with the best checkpoint picked by the refinement stage.

At the testing time, the multiple scale testing and horizontal flip testing are applied. As for the Frequency Weighted (FW) model ensemble method, we obtain the frequency statistics of all the classes in the training dataset of MiniCity, and find out all the classes with frequency less than 1%. For those classes, we assign a higher weights 3.0 in the logits of the OHEM model and seg-Augmix model.

**VIPrior Challenge Results** The performance of our entry to the challenge is presented in the Table 8. It achieved the 3rd position in the Semantic Segmentation track of the challenge. The number is obtained by a FW ensemble of HRNetV2-W48 + OCR, HRNetV2-W48 + OCR + OHEM, and HRNetV2-W48 + OCR + seg-Augmix, and HRNetV2-W44 + OCR models.

**Ablation Study** The performance of HRNetV2-W48 + OCR, HRNetV2-W48 + OCR + OHEM, and HRNetV2-W48 + OCR + seg-Augmix models are presented in the Table 7, on the validation dataset of MiniCity. We can see that the proposed seg-Augmix model can improve the performance of the model without it. Besides, the proposed Frequency Weighted (FW) model ensemble method also improve upon the average model ensemble.

# 5    Conclusions and Discussions

This paper presented diversification of the models, the data and the test samples for data efficient image understanding problems, namely image classification and semantic segmentation. We observed several interesting phenomena that can be further investigated. A simple combination of Augmix and Mixup can boost the performance for image classification. The potential reason might be that Augmix focuses more on within-class data diversification, while Mixup focuses more on between-class diversification. The emperical results also demonstrated the effectiveness of the proposed seg-augmix for semantic segmentation tasks. The

Frequency Weighted (FW) model ensemble method utilized the property of the models in the model ensemble and further improved the semantic segmentation performance. In the future, we plan to extend Augmix + Mixup to semantic segmentation task for performance improvement.

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
