# OpenReview forum: "Diversification is All You Need: Towards Data Efficient Image Understanding"
_thecvf.com/ECCV/2020/Workshop/VIPriors — Submitted to VIPriors_

### Official Review · AnonReviewer2 · 2020-07-21
**Effective method but lack of novelty and out of scope**

**Confidence:** 4
**Rating:** 5

**Review:**

#### 1. [Summary] In 2-3 sentences, describe the key ideas, experiments, and their significance.
The paper proposes a combination of extensive data augmentation and model ensembling to train deep neural networks in a data efficient manner. Experiments are performed on small-size classification and segmentation datasets and performance improvements are shown.

#### 2. [Strengths] What are the strengths of the paper? Clearly explain why these aspects of the paper are valuable.
* Clarity: the paper is clear and easy to read.
* Effectiveness: the method seems effective on small-size datasets.
* Reproducibility: the experiments are well documented and include relevant hyperparameters.
* Experiments: the effectiveness of the method is empirically demonstrated through ablation studies.

#### 3. [Weaknesses] What are the weaknesses of the paper? Clearly explain why these aspects of the paper are weak.
* The papers mostly makes use of existing data augmentation techniques and therefore lacks novelty. The only seemingly novel contribution is the Frequency Weighted (FW) model ensemble method for dealing with the imbalanced class distribution in the semantic segmentation experiments.
* (l. 334-359) The paper claims that FW model ensembling is beneficial for low frequency classes but does not support this claim with empirical results. The authors should have included a comparison of per-class IoU scores between the baseline method and the proposed method.
* (l. 147-149) "...mixup extends the training distribution by incorporating the prior knowledge that linear interpolations of feature vectors should lead to linear interpolations of the associated targets."
The argument of the proposed method for incorporating prior knowledge is fairly weak and is related to existing work (Mixup). The paper therefore seems to fall outside of the intended scope of this workshop.

#### 4. [Overall rating] Paper rating
* 5: Marginally below acceptence threshold

#### 5. [Justification of rating] Please explain how the strengths and weaknesses aforementioned were weighed in for the rating.
Despite the effectiveness of the data augmentation and model ensembling techniques, the proposed method is mainly not novel and the argument for incorporating prior knowledge is weak.

#### 6. [Detailed comments] Additional comments regarding the paper (e.g. typos or other possible improvements you would like to see for the camera-ready version of the paper, if any.)
* A table including per-class IoU scores and class occurances should be included for the segmentation experiment.
* It would be nice to see example images for the proposed data augmentation method for image classification as well.

---

### Official Review · AnonReviewer1 · 2020-07-27
**Diversification is All You Need: Towards Data Efficient Image Understanding**

**Confidence:** 5
**Rating:** 5

**Review:**

#### 1. [Summary] In 2-3 sentences, describe the key ideas, experiments, and their significance.
The paper shows an ensemble of multiple models, which are trained with various augmentation techniques, improves the performance of image classification and segmentation tasks.

#### 2. [Strengths] What are the strengths of the paper? Clearly explain why these aspects of the paper are valuable.
- Performance on classification and segmentation tasks

#### 3. [Weaknesses] What are the weaknesses of the paper? Clearly explain why these aspects of the paper are weak.
- There is no proper Related Works section.
- The structure of the paper is hard to follow and there are jumps from a topic to another.
- A lot of missing citations

#### 4. [Overall rating] Paper rating
5

#### 5. [Justification of rating] Please explain how the strengths and weaknesses aforementioned were weighed in for the rating.

The paper is more technical report than an academic paper.

#### 6. [Detailed comments] Additional comments regarding the paper (e.g. typos or other possible improvements you would like to see for the camera-ready version of the paper, if any.)
- Missing citations:
	> L.37-39: '..tasks..'
	> L.45: 'Methods' (only one method)
	> L.56: 'augmentation methods'
	> L.73: 'recently several works..'
	> L.78: 'frequency weighted model ensemble'
	> L.361: '..widely know..'
	> L.367: 'Conventional semantic segmentation models..'
	> L.400: 'RMSProb.'
- L.217: The second part of formula is not explained.
- L.267: No need for "Please.."
- L.298: What is "m" in the formula and Fig 2?
- L.313: "TTA" can be given in paranthesis.
- L.337: It is not clear if the sentence is about the segmentation or both classification and segmentation.
- L.355, 356, 368 etc: Future tense can be changed as simple present tense.
- L.343, 344, 347, 352: Frequency can be misunderstood, instead, frequent or occuring etc can be used.
- L.374: HRNetV2 and Object Contextual Representation methods should be in Related works section.
- L.399: Link can be footnote.
- L.478: 'Tab.'. There should be a consistency about refering the tables. In some cases they are as 'Table' and for other cases as 'Tab.'
- L.516: Did the authors train the segmentation networks 4840 epochs or it is a typo?
- L.579: "We observed several interesting phenomena.." It is not clear that what are those findings.
- Please remove intermediate horizontal lines in the tables (booktabs latex)

---

### Decision · Program_Chairs · 2020-07-29

**Decision:**

Reject

**Comment:**

After considering the reviews and further discussion, we do not find sufficient cause to overturn the recommendation of the reviewers.